# Evaluation of Blood Soluble CD26 as a Complementary Biomarker for Colorectal Cancer Screening Programs

**DOI:** 10.3390/cancers14194563

**Published:** 2022-09-20

**Authors:** Loretta De Chiara, Leticia Barcia-Castro, María Gallardo-Gómez, María Páez de la Cadena, Vicenta S. Martínez-Zorzano, Francisco J. Rodríguez-Berrocal, Luis Bujanda, Ane Etxart, Antoni Castells, Francesc Balaguer, Rodrigo Jover, Joaquín Cubiella, Oscar J. Cordero

**Affiliations:** 1Department of Biochemistry, Genetics and Immunology, Universidade de Vigo, 36210 Vigo, Spain; 2CINBIO, Universidade de Vigo, 36210 Vigo, Spain; 3Galicia Sur Health Research Institute (IIS Galicia Sur), SERGAS-UVIGO, 36213 Vigo, Spain; 4Department of Gastroenterology, Biodonostia Health Research Institute, Centro de Investigación Biomédica en Red de Enfermedades Hepáticas y Digestivas (CIBERehd), Universidad del País Vasco (UPV/EHU), 20014 San Sebastián, Spain; 5Department of Surgery, Hospital Universitario Donostia, 20014 San Sebastián, Spain; 6Gastroenterology Department, Hospital Clinic, IDIBAPS, CIBERehd, University of Barcelona, 08036 Barcelona, Spain; 7Department of Gastroenterology, Hospital General Universitario de Alicante, 03010 Alicante, Spain; 8Department of Gastroenterology, Complexo Hospitalario Universitario de Ourense, CIBERehd, 32005 Ourense, Spain; 9Department of Biochemistry and Molecular Biology, CIBUS Building, University of Santiago de Compostela, 15782 Santiago de Compostela, Spain

**Keywords:** colorectal cancer, advanced adenomas, FIT, endoscopy, blood test, soluble CD26, DPP4

## Abstract

**Simple Summary:**

Many countries have implemented, or are implementing, general age-based screening programs to reduce mortality due to colorectal cancer. Fecal hemoglobin immunodetection (FIT) in combination with endoscopy has already been relatively successful in achieving this goal. However, there are issues that can be improved in relation to participation rates and reduction in false positives. We studied whether the blood biomarker soluble-CD26 (sCD26), a glycoprotein with dipeptidyl peptidase enzyme activity (DPP4), could help in the early diagnosis of colorectal cancer and advanced adenomas in combination with FIT, also reducing false positives. We propose a sequential testing strategy for FIT positive individuals, offering an alternative blood test with our biomarker for a confirmation prior to colonoscopy in the short term.

**Abstract:**

Fecal hemoglobin immunodetection (FIT) in combination with endoscopy has been implemented to reduce mortality from colorectal cancer (CRC), although there are issues that can be improved in relation to participation rates. We studied whether the blood biomarker soluble-CD26 (sCD26), related at least in part to the immune system and inflammation, and/or its dipeptidyl peptidase enzyme activity (DPP4), could help reduce false positives. In a cohort of 1703 individuals who underwent colonoscopy and had a serum sample, sCD26 and DPP4 activity showed statistically significant differences regarding sex and age. According to the colonoscopy findings, sCD26 and DPP4 activity progressively decreased in advanced adenomas and CRC, with statistically significant differences, even between both groups; 918 of them had a FIT result (*n* = 596 positive cases) with approximately 70% of these (*n* = 412) false positives. With cut-offs of 440 ng/mL for sCD26, 42 mU/mL for DPP4, and 11 ng/mU for their ratio, the combined information of the three biomarkers (at least positive for one biomarker) identified almost all advanced adenomas and CRC cases in the FIT cohort with approximately half of the false positives compared to FIT. A sequential testing strategy with FIT and our blood biomarker test is proposed.

## 1. Introduction

Colorectal cancer (CRC) is the most prolific cause of cancer-related deaths in the world [1]. Many countries have implemented, or are in the process of implementing, age-based general screening programs for the early detection of CRC since mortality is highly dependent on cancer stage at diagnosis [2,3]. 

These programs have already shown their effectiveness [4,5,6,7,8]. Colonoscopy has not only therapeutic but also preventive effects through the removal of precancerous or early malignant lesions, i.e., polyps, reducing CRC incidence. However, colonoscopy as a general tool for population-wide screening faces many logistic and economic-related inconveniences, as well as low participation rates due to its invasiveness [6,7]. Strategies for the detection of pre-clinical cancer in a screening context rely on the development of optimal biomarkers that can select individuals for colonoscopy, which is the gold standard. A major success was the evolution of fecal tests, currently the immunodetection of hemoglobin, followed by endoscopy [3,7,8]. Notwithstanding, stool-based tests for screening suffer from low adherence due to people’s reservations related to stool collection, handling, and storage [9], together with false positives in individuals with hemorrhoids, among others. In addition, the sensitivity of the fecal immunochemical test (FIT) is moderate for the detection of premalignant advanced adenomas (AA) [10]. Therefore, ongoing research is still focused on the development of minimally invasive, resource-effective, and robotized blood-based tests to increase participation in population-based screening programs [2,6,11].

We have recently reviewed the most robust and commonly used high-throughput approaches for CRC blood-based biomarker discovery in the fields of genomics, transcriptomics, epigenomics, proteomics, and metabolomics [12], including the FDA-approved test Septin 9 DNA methylation, CancerSEEK, Guardant360 CDx, or FoundationOne Liquid CDx. However, they have not achieved the desired accuracy to be included in screening programs [13], perhaps because studies are commonly based on samples collected from clinical settings of patients with existing CRC [14]. New studies are using screening settings where samples come from individuals with colorectal adenomas and polyps, but not adenocarcinomas [11,14], or using panels of inflammatory biomarkers [15] as inflammation is a hallmark of cancer and an etiological driver of CRC [16], often adding data mining from meta-analyses.

One of the biomarkers discovered in CRC patients is sCD26, the soluble form of cell surface CD26, which is present in many tissues, related at least in part to the immune system [17,18]. It might be a confounding biomarker for advanced CRC stages since cell surface CD26 is a biomarker of some metastatic cancer stem cells (CSC) [19], and serum sCD26 levels were found enhanced in CRC advanced tumor stages [20,21]. However, sCD26 levels are impaired in the early stages. We have reported encouraging sensitivity and specificity [22], including a relationship between sCD26 and the grade of dysplasia and the presence of AA [23]. We hypothesize, at present, that some change in CD26+ immune subsets might be responsible for this sCD26 impairment [24,25,26]. Several studies in different cancers have found low correlations of sCD26 levels with its enzymatic activity dipeptidyl peptidase 4 (DPP4, EC 3.4.14.5) due to some referenced factors [27,28,29,30]. The aim of this study was to determine if sCD26 and its enzymatic activity can complement the performance of FIT for the early detection of CRC and AA, analyzing these biomarkers in cohorts of cancer and average-risk populations derived from the national screening program.

## 2. Materials and Methods

### 2.1. Study Population 

A prospective, controlled, double-blinded study was designed. The cohort included a total of 1703 patients recruited from four Spanish Hospitals conducting screening programs: 436 patients from Complexo Hospitalario Universitario de Ourense (Ourense), 601 from Hospital de Donostia (San Sebastián), 480 from Hospital Clínic de Barcelona (Barcelona), and 186 from Hospital General Universitario de Alicante (Alicante). Details of the complete cohort are provided in Table 1. All recruited individuals underwent a colonoscopy and had a blood sample taken before colonoscopy. Additionally, some of the individuals from the cohort also had a FIT test. Not all patients had the FIT because some did not come from the screening programs but from receiving health care for other reasons (symptoms, risk of cancer, etc.) in the Digestive Services. Exclusion criteria included a personal history of CRC, digestive cancer or inflammatory bowel disease, severe synchronic illness, or a previous colectomy.

The study was conducted according to the clinical and ethical principles of the Spanish Government and the Declaration of Helsinki and was approved by the Ethics Committee for Clinical Research of Galicia (2018/008). Informed consent was obtained from each individual and anonymity was warranted.

### 2.2. Colonoscopy

The colonoscopy was performed by experienced endoscopists following the recommendations of the Spanish guidelines on the quality of colonoscopy in CRC screening [31]. 

Individuals were classified based on the diagnosis of colonoscopy and according to the most advanced lesion as: no neoplasia (NN; including no colorectal findings (NCF) and benign pathologies), non-advanced adenomas (NAA), AA, and CRC. AA are defined as adenomas ≥ 1 cm, with villous components or high-grade dysplasia. Cancer was classified according to the AJCC staging system [32,33]. Advanced neoplasia (AN) was defined as AA or invasive cancer. The location of the lesions was classified as ‘proximal’ when located only proximal to the splenic flexure of the colon, and ‘distal’ when lesions were found only in the distal colon or in both distal and proximal colon.

### 2.3. Blood Samples and Measurement of sCD26, DPP4 and Total Protein

Blood was drawn 1 week before the endoscopic procedure and coagulated at room temperature for 20 min and centrifuged at 2000 g for 15 min. The serum samples obtained were stored at −20 °C.

The sCD26 concentration was measured in duplicate with the Human sCD26 platinum ELISA kit (eBioscience; Vienna, Austria) according to the manufacturer’s instructions. Colorimetric quantification was performed with a microplate reader (model 550; Bio-Rad; Hercules, CA, USA) at a wavelength of 450/570 nm.

DPP4 enzyme activity was measured in duplicate in 96-well culture plates with substrate Gly-Pro-p-nitroanilide (1 mM, Merck Sigma Aldrich, Burlington, MA, USA) in reaction mixtures (100 μL) containing serum samples (10 μL) and 50 mM of Tris-HCl, pH 8.0. At time 0 (blank), and after incubation of the plates for 30 min, the hydrolysis of the substrate was monitored at a wavelength of 415 nm in the Bio-Rad microplate reader. The results were obtained by comparison with the standard curve for p-nitroaniline (Merck Sigma Aldrich, Burlington, MA, USA) and the enzymatic activity was expressed as mUL^−1^. 

Total serum protein was quantified using the classical Biuret test to determine sCD26-specific enzymatic activity.

### 2.4. Stool Samples and FIT 

A fecal quantitative immunochemical test was available from a subset of the patients included in the cohort. Typically, a stool sample was collected the week prior to the colonoscopy without specific diet or medication restrictions. The test used in all hospitals was the automated OC-Sensor (Eiken Chemical, Tokyo, Japan). The chosen FIT cut-off, 100 ng/mL, is the default setting defined by the manufacturer as well as the standard used in Spanish screening programs.

### 2.5. Data Analysis

The data were included in a specifically designed database. Two or multiple-group comparisons were performed using the Student’s *t*-test and ANOVA, respectively. The ability of sCD26, DPP4, and the sCD26/DPP4 ratio to separate no neoplasia from advanced neoplasia was determined setting cut-offs to maximize the detection of AA and CRC. A test was considered positive when the levels of each marker were lower than the cut-off, while a test was negative when the levels were higher. 

Statistical analyses were performed with SPSS software (v.20.0). *p*-values ≤ 0.05 were considered statistically significant.

## 3. Results

### 3.1. Baseline Characteristics of the Study Population

The study included 1703 individuals (Table 1) that underwent a colonoscopy and had a serum sample. Individuals consisted of 853 men (50.09%) and 850 women (49.91%) with ages ranging from 29 to 92 years. 

According to the colonoscopy findings, patients were classified as follows: 1082 with NN comprising NCF (*n* = 344) and benign pathologies that included hemorrhoids (*n* = 119), diverticula (*n* = 97), inflammatory or hyperplastic polyps (*n* = 46), and other minor findings (*n* = 24), such as angiodysplasia (*n* = 6), rectitis/proctitis/unspecific inflammation (*n* = 5), melanosis coli (*n* = 4), and parasites (*n* = 2), among others; 452 NN cases were NAA. There were 372 AA and 249 CRC cases (73 stage I, 75 stage II, 77 stage III, 21 stage IV, and 3 of unknown stage).

sCD26 and DPP4 levels were measured in 1666 and 1174 individuals, respectively, while a total of 1137 individuals had both measurements. Protein serum concentration was quantified in 1174 individuals. Regarding FIT, 917 (53.85%) of the 1703 individuals included in the cohort had a FIT result. Individuals with a FIT result consisted of 455 men (49.6%) and 462 women (50.4%), with ages ranging from 29 to 82 years.

### 3.2. Serum sCD26 and DPP4 Enzymatic Levels in the Study Population

We first analyzed the correlations between soluble CD26 levels, DPP4 activity, and total protein levels in the cohort. The correlation found between sCD26 and DPP4 activity was moderate (Pearson test, *r* = 0.603, *p*-value < 0.001), while no association was found between each of these measurements and total protein concentration (Pearson test, *r* = 0.046 and r= 0.065, respectively). sCD26 and DPP4 also correlated very slightly with the specific activity (Pearson test, *r* = 0.190 and 0.260, *p*-values <0.001). Therefore, all these parameters and their ratios were initially tested.

Three age groups were established (≤49, 50–59, and ≥60 years; Table 2). The ANOVA test (*p*-value < 0.001) showed differences in sCD26 and DPP4 activity comparing age groups. Since sex differences were found, the analysis according to age groups was performed separating women and men, finding a complex pattern (Table 2). In general, there is a gradual decrease in DPP4 activity and sCD26 levels with age in both sexes. To note, in the small group of youngest women, the sCD26/DPP4 ratio was considerably lower, i.e., the diminution in sCD26 levels is greater than that of DPP4. 

### 3.3. Serum sCD26 and DPP4 Levels in the Study Population According to the Colonoscopy Findings

Serum sCD26 levels together with DPP4 activity, total protein concentration, specific DPP4 activity and sCD26/DPP4 ratio according to the colonoscopy findings are shown in Table 3. There were heterogenous sCD26 levels in the NN subgroups. However, sCD26 progressively decreased in AA and CRC (ANOVA test, *p*-value < 0.001), with statistically significant differences even between AA and CRC (Student’s *t*-test, *p*-value < 0.001). Similar results were found for DPP4 activity comparing no neoplasia, AA, and CRC (ANOVA test, *p*-value < 0.001). 

In both cases, although the NN and NAA means were similar, the latter showed statistical differences both with NCF and advanced neoplasia. 

For total protein levels, the mean of the groups was comparable, although the distribution was not normal in all the groups. For the other two variables, specific activity and ratio sCD26/DPP4, most but not all of the statistically significant differences were maintained. Given the differences observed in men and women, we analyzed all the parameters in each sex group separately (Appendix A). Mean sCD26 levels in men and women with NN were 496.27 and 536.12 ng/mL, respectively, whereas in AN these were 367.94 and 402.64 ng/mL, respectively. DPP4 activity in men and women was 40.30 and 45.98 mU/mL, respectively, in NN, and 32.25 and 36.82 mU/mL in AN. The differences mentioned before were largely maintained except for the sCD26/DPP4 ratio, with statistically significant differences found only in males (Appendix A) but not in females (Appendix A), which showed comparable values in NN and CRC.

### 3.4. Evaluation of Serum sCD26 and DPP4 Levels in the NN Group According to the Colonoscopy Findings

Because of the heterogeneity found in this group and to seek circumstances of false positives for these biomarkers, we also analyzed sCD26 levels, DPP4 activity, total protein concentration, specific DPP4 activity and sCD26/DPP4 ratio in the NN subgroups. For the first time, statistically significant differences for sCD26 and DPP4 were found among the subgroups included in this collective. Comparisons between NCF and the rest of the NN subgroups are also shown in Table 3. Statistically significant differences were found in sCD26 for hemorrhoids, diverticula, polyps, and NAA. For DPP4, differences were observed only for polyps and NAA, while for the sCD26/DPP4 ratio, diverticula resulted in the only subgroup with differences. Curiously, when the analysis was performed by sex (Appendix A), only sCD26 levels (but not DPP4) were statistically different in men with diverticula (mainly) and NAA, whereas in women, differences were only found in the hemorrhoids group. Note the different statistical results between both markers. Among individuals with other minor findings, those few (*n* = 6) with angiodysplasia showed higher sCD26 and DPP4 levels.

### 3.5. Preliminary Diagnostic Accuracy of sCD26, DPP4, and Their Ratio in the FIT Cohort

A total of 917 individuals (53.85%) from the cohort underwent FIT, of which 596 had a positive result (64.99%), while 321 had a negative test (35.01%). In this sub-cohort, the proportion of positive and negative tests are far from that of a screening context, thus, evaluation of sensitivity or specificity would be biased and was not estimated. 

In a typical FIT-based screening program there are two main problems: the undetected AA (false negatives for FIT) and the over-diagnosed cases (false positives for FIT). In this sub-cohort, approximately 70% of FIT positive cases (*n* = 412) were false positives.

Considering that the objective of this work is not a comparison with FIT but to evaluate our biomarkers as complementary tests, we analyzed sCD26, DPP4, and their ratio among FIT positives and negatives (Figure 1A,B). As preliminary cut-offs for our biomarkers, we used the mean values observed for AA in women, which reported higher means than men. This criterion was chosen to detect the majority of AA and CRC cases in both sexes. According to the cut-offs used, individuals were sCD26 positive when levels were ≤440 ng/mL, DPP4 positives when ≤42 mU/mL, and sCD26/DPP4 (ratio) positives when ≤11 ng/mU. 

Among FIT positives (Figure 1B), sCD26 was detected in 15/21 CRC cases (71.43%), DPP4 in 16/18 (88.89%; activity was not measured in 3 cases), and the sCD26/DPP4 ratio in 10/18 (55.56%). Interestingly, with the combined information of the three biomarkers (positive for at least one of them), only one case was not detected (20/21; 95.24%). In this CRC case (woman, stage I), only sCD26 was measured, resulting in 468.20 ng/mL. Since DPP4 activity was not measured in this case, we are not able to know if its measurement would have contributed to the detection of cancer, in particular, because with both measures, the ratios of sCD26/DPP4 showed a wide range (from 4 to 22) and many were outliers.

Most AA were detected as well in this FIT positive sub-cohort (Figure 1B). Concretely, 111 out of 155 (71.61%) were positive for sCD26, 82 out of 108 (75.93%) for DPP4 activity, and 65 out of 100 (65.00%) for the ratio. When considering at least one of the biomarkers positive, 131 out of 163 AA (80.37%) were detected. The 32 undetected AA were as follows: 7 were negative for sCD26, DPP4, and the ratio; 6 were negative for DPP4 (sCD26 was not measured); 19 were negative for sCD26 (DPP4 was not measured). 

In the FIT negative cohort, only a few cases had DPP4 values, therefore, we were unable to combine the information of the biomarkers. However, of the 23 AA cases, 17 (73.91%) were sCD26 positive, and of the 6 FIT negative AA, 3 showed high levels of the marker. These results altogether support that this protein is a good biomarker candidate for a multiplex blood test. 

As stated above, an important inconvenience of FIT-based screening is false positives. In this FIT positive sub-cohort (Figure 1A), 116/198 individuals with no neoplasia (excluding NAA) were correctly classified as negative according to sCD26 (58.59%), 87/148 (58.78%) for DPP4, and 71/146 (48.63%) for the sCD26/DPP4 ratio. For all markers, approximately half of the false positives in the NN group without NAA belonged to the diverticula and hemorrhoids groups. Because of the differences between sex mentioned above, this latter analysis was separated by sex, finding an interesting difference for the ratio, 22 false positives in women and 53 in men. In relation to NAA, among the FIT positive cases, 105/211 (49.76%) were classified as sCD26 negative, 87/180 (48.33%) as DDP4 negative, and 83/179 (46.37%) as negative for the sCD26/DPP4 ratio. 

### 3.6. Proposal for the Integration of Our Biomarkers in FIT-Based Colorectal Cancer Screening

According to our results, what seems closer to a possible clinical use of our biomarkers is the reduction in the number of colonoscopies among individuals with a false positive FIT result. This issue is important not only because of the cost of the endoscopy but also the discomfort and risks related to the procedure for the patient. 

We first tested a sequential strategy starting with a FIT test, as is performed nowadays, followed by the inclusion of our biomarkers in the sub-cohort of FIT+ cases with all measures (*n* = 443). In this group there were 325 false positives (no AA or CRC). Furthermore, 212 patients tested sCD26 + (143 FP), but there were 5 CRC and 25 AA not detected. With the DPP4 activity and ratio data, using the whole group or alternatively the subgroup of CD26 negative cases, all CRC cases and most AA were detected. However, only 68 FP were negative for the three CD26 measures, i.e., not many colonoscopies would be avoided (20.92% in comparison to when only FIT is used).

Consequently, Figure 2 summarizes our proposal, that, instead of the indication of a colonoscopy based only on a positive FIT, a blood sCD26 test will be offered to these FIT+ individuals. Double-positive individuals, which is FIT+ and sCD26 +, will be submitted to colonoscopy, while FIT- and sCD26/DPP4—will be tested again in 2 years with FIT. 

Positivity is questioned in the case of a blood negative test (*n* = 231). The additional step offered to them is the repetition of the FIT and our sCD26/DPP4 test in the weeks following the first test, expecting that false FIT positives might return to negativity, or sCD26/DPP4 negatives to positivity. These persons avoiding the colonoscopy must be studied under this proposed validation phase.

## 4. Discussion

A recent meta-analysis evaluating the performance of FIT for screening reported a sensitivity for CRC and AA from 71–91% and 25–40%, respectively, and a specificity from 90–95% [34]. Some of the differences are related to the type of FIT test, but mainly to the threshold used. In our cohort, coming from the Spanish Screening Program where colonoscopies are funded by the public health service, the threshold established is 100 ng/mL, which allows higher sensitivity accompanied by lower specificity. The real performance of FIT in our study could not be estimated as a small proportion of FIT negatives performed a colonoscopy. Here, all CRC cases in the sub-cohort with FIT were FIT+ [35], however, a relevant number of FIT positives underwent a colonoscopy showing no colorectal findings or benign pathologies. Moreover, FIT has a low detection capacity for premalignant AA, detecting distal lesions better than proximal ones [36] as well as a reduced capacity to detect serrated lesions [37]. 

Many results highlight the challenge of developing biomarkers that are effective in the asymptomatic, pre-diagnostic window of opportunity for the early detection of CRC. Additionally, it seems that a single biomarker or a small panel may not render the desired precision for screening. However, we and other groups have consistently shown that serum sCD26 levels alone is quite an informative biomarker for early CRC and AA [24,25,26,27,28,29,30,32,38]. Again, the results in this work support its candidature for a multiplex blood test [38], as our biomarkers were able to detect all CRC cases, above the 70% of AA that were FIT negative and 93% of AA that were FIT positives.

Interestingly, a very recent work investigating expression levels of 15 pluripotency-associated genes for the detection of dysregulation of CSC or stem cell (SC) markers (including CD26) in a rectal swab specimen, has shown that the downregulation of a couple of genes, CD26 together with Oct4, can be a new promising screening method for high-risk patients [37]. In this work, the authors did not answer why a marker of CSCs was found downregulated. The gene CD26 should be overexpressed if we look for the metastatic potential of CSCs in tumors [19,39], as was confirmed for the serum protein levels, DPP4 enzymatic activity, and gene transcription in the tumor [20,21]. However, in the early progression steps of this and other cancers, sCD26/DPP4 levels are lower than in healthy donors [17,18,22,23,24,27,28,29]. That is an impairment, along with the fact that CD26 is not a tumor-associated antigen (TAA) has overlooked this marker. However, our work can be related to the new studies using screening settings from individuals with adenomas and polyps or using panels of inflammatory biomarkers [11,14,15]. In fact, most data relate the changes of sCD26 levels with the immune system [18,24,25,26,40,41] with probable impact, as deduced from pre-clinical models, in DPP4 substrates such as certain chemokines [42,43]. This biomarker, therefore, is in line with a recently implemented Systemic Immune-Inflammation Index for CRC [44]. In this sense, the recently developed Fecal DPP4 test seems rather informative of the DPP4 on the apical PM of epithelial cells [45]. 

On the contrary, this lack of specificity may prevent its use as a single biomarker in a cancer screening context. In fact, some of the false positives for AA/CRC can be indicating other tumors or conditions where their levels are also reduced [18].

Therefore, to explore a clinical use of our marker, we focused on its performance as complementary to FIT, to avoid FIT false positives. We could have opted to validate the cut-off of our previous paper for sCD26, 410 ng/mL [32], but the cut-offs used in this study were chosen to detect most AA (increased sensitivity). Although, unfortunately, some DPP4 cases were lost, it seems that our marker (protein, enzymatic activity, and their ratio) can identify almost the same CRC and AA cases detected by FIT, but with a smaller number of false positives, and less than half in the group of NCF. Among these, around half belonged to the hemorrhoids and diverticula groups. Due to the sex differences found in the sCD26/DPP4 ratio analysis, it is likely that the number of false positives could still be reduced by having those measures. 

The most relevant benefit from the inclusion of our blood markers in screening is the reduction in the number of colonoscopies performed to false positive individuals (FIT+). The estimated cost per patient for performing a sCD26 test is EUR 4 and EUR 1 for DPP4, in addition to the cost of laboratory personnel (at least 20-fold cheaper than a colonoscopy, including polypectomy and pathology costs and not considering the cost of the establishment of a whole population screening program by the public health system). While the cost of a colonoscopy is considerably greater than our blood tests, importantly, unnecessary invasive colonoscopies will be avoided among this subgroup of FIT false positive patients. The strategy proposed, therefore, translates into savings for health systems. Additionally, a decrease in the rate of false positive results when combining FIT and the blood test will increase screening participation rates since both general practitioners and patients will be more receptive to national screening programs.

In this work, DPP4 activity was measured in parallel because we had found that the sCD26 levels show low correlations with its enzymatic activity [27,28,29]. The ratios between both values as measured in this work show again an important dysregulation. Early on, it was found that exosomes of many origins contain plasma membrane CD26 [46]. Very recent studies have reported exosomes enriched in CD26/DPP4 in a colorectal cancer context and related to angiogenesis [30,47]. Probably, most exosomes did not remain bound to the anti-CD26 antibody of the ELISA test while its DPP4 activity was measured in the serum. The finding that patients with angiodysplasia showed high DPP4 levels could be related to this fact.

However, there are additional explanations. For example, (i) some circulating proteins other than sCD26 display DPP4 activity [27,28,29]; (ii) the inhibitory effect of hypersialylation on DPP4 activity, strongly enhanced in elderly individuals and supported by our data on individuals older than 50 years [rev. in 18] (although this fact would not affect the age-based screening programs and the effect was different in both sexes); (iii) the presence of some proteins that may bind to CD26 may regulate its DPP4 activity [18,22,23,24,27,28,29]. Some of these have been found altered in cancer, for example, FAP (seprase) or glypican-3 [48,49]. Likewise, it has been reported that hypoxia regulates DPP4 MMP-mediated proteolytic inactivation and shedding from ovarian cancer cells [50]. Last, an effect of the anti-CD26 autoantibodies [28], which can bind to sCD26 cannot be discarded.

Another remarkable fact is the difference in sex. Our study clearly supports that the higher levels of DPP4 activity in women, described numerous times [18], are based on the higher levels of sCD26, and this is particularly seen in individuals older than 50 years. Although in this cohort the cases in the youngest group are few, the trend was different not only because the sCD26 levels were higher in men but the DPP4 activity was also similar, as observed in other cohorts of different ages [28]. This can be explained by an associated factor in women avoiding the sCD26 capture in the ELISA assay, or alternatively, enhancing its enzymatic activity [28]. Interestingly, sex differences in the behavior of DPP4 were related to the energy metabolism [51] and also to CD26 expression in the immune system [52]. As mentioned, most sCD26 is apparently originated from immune cells in a non-pathological condition [18,26]. Still, at present, it is difficult to understand the differences seen in the hemorrhoids and diverticula groups, where sCD26 but not DPP4 levels were shown for the first time to be statistically different, with lower levels in women with hemorrhoids and in men with diverticula.

Although most CRC cases are identified by our biomarker (sCD26 and DPP4), the number of cases rescued from colonoscopy would be scarce using the complete blood test. Despite the large overall size of the study, the main limitation was due to the lack of enough serum from some patients to perform some determinations. The low number of FIT negatives is explained because the majority of colonoscopies came from the screening programs (in other cases there was no information). Because of this, our proposal for a sequential screening strategy is the complementary blood analysis. This would lead to a second FIT and complete blood test to avoid the loss of cases; in this way our proposal could be validated in a screening context.

Additional information to this strategy could be introduced if necessary. For example, a recent work [53] suggests a novel method to measure DPP4-dependent biologically relevant chemokine CXCL10 (one of its substrates) proteoforms in clinical samples. 

## 5. Conclusions

We studied whether the blood biomarker soluble-CD26 (sCD26), a glycoprotein with dipeptidyl peptidase enzyme activity (DPP4), could help in the early diagnosis of colorectal cancer and advanced adenomas in combination with FIT, and/or reducing false positives. 

In spite of some biological interesting findings such as the sex differences and the low levels in diverticula or hemorrhoids, we propose a sequential testing strategy for FIT positive individuals having a confirmation prior to colonoscopy with our blood test in the short term. 

## Figures and Tables

**Figure 1 cancers-14-04563-f001:**
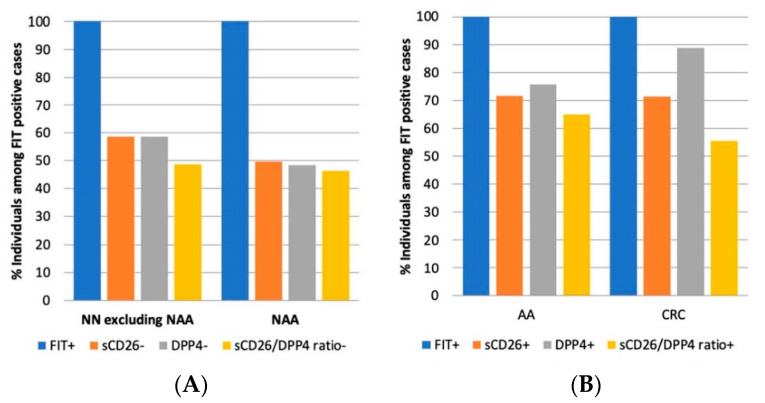
Frequencies of positive and negative cases from the blood test among FIT positives with no neoplasia (**A**) and advanced neoplasia (**B**). (**A**), proportion of negative cases for sCD26, DDP4 and sCD26/DDP4 among FIT positives with no neoplasia; (**B**), proportion of positive cases for sCD26, DDP4 and sCD26/DDP4 ratio among FIT positives with advanced neoplasia.

**Figure 2 cancers-14-04563-f002:**
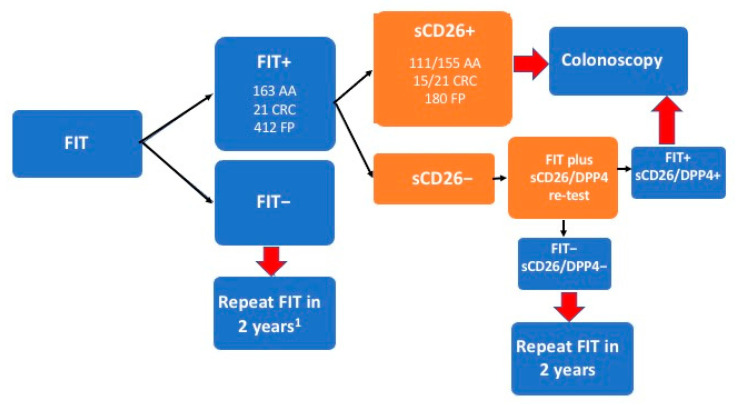
Proposal for the integration of our blood test in FIT-based colorectal cancer screening. ^1^ sCD26/DPP4 blood test will not be offered to FIT negatives; instead, they will repeat FIT in 2 years.

**Table 1 cancers-14-04563-t001:** Epidemiological and clinical characteristics of the patients recruited per hospital.

CENTER	Complexo Hospitalario Universitario de Ourense	Hospital Donostia	Hospital Clínic de Barcelona	Hospital General Universitario de Alicante	Total Patients Recruited
**Number of patients recruited**	436	601	480	186	1703
**Sex (% men)**	47.7%	52.2%	46.3%	58.6%	50.1%
**Age range and mean**	29–87 (58.2)	41–92 (64.5)	49–70 (59.7)	48–85 (63.6)	29–92 (61.5)
**Ne neoplasia**	**333**	**334**	**373**	**42**	**1082**
NCF	116	148	73	7	344
hemorrhoids	58	7	46	8	119
diverticula	43	6	43	5	97
polyps ^1^	33	2	9	2	46
others ^2^	6	1	16	1	24
NAA	77	170	186	19	452
**Advanced adenomas**	**72**	**131**	**82**	**87**	**372**
distal	47	83	39	68	237
proximal	25	48	43	19	135
**Colorectal cancer**	**31**	**136**	**25**	**57**	**249**
Stage I	4	32	18	19	73
Stage II	4	52	7	12	75
Stage III	19	39		19	77
Stage IV	4	13		4	21
Unknown stage				3	3

NCF: no colorectal findings; ^1^ inflammatory and hyperplasic polyps; ^2^ others include angiodysplasia, rectitis and melanosis coli, among others; NAA: non-advanced adenomas.

**Table 2 cancers-14-04563-t002:** Showing sCD26 and the other measurements regarding sex and age. Statistically significant differences (Student’s *t*-test, *p*-value < 0.001) between sex were found for sCD26 and DPP4 activity, both higher in women, but no differences were detected for the other parameters.

Variable	sCD26 Mean ± SD (ng/mL) N *p*-Value			DPP4 Mean ± SD (mU/mL) N *p*-Value			Total Protein Mean ± SD (mg/mL) N *p*-Value			Sp. Act (DPP4/Prot) Mean ± SD (mU/mg) N *p*-Value			sCD26/DPP4 Ratio Mean ± SD (ng/mU) N *p*-Value		
**Sex** Women Men	840 826	498.30 ± 189.29 440.96 ± 192.04	**<0.001** ^1^	573 601	42.77 ± 12.39 36.30 ± 12.51	**<0.001** ^1^	573 601	76.56 ± 13.06 76.50 ± 14.16	0.939 ^1^	573 601	0.57 ± 0.18 0.51 ± 0.73	0.076 ^1^	563 574	11.79 ± 5.37 12.18 ± 5.39	0.219 ^1^
**Age (years)**															
≤49	126	503.68 ± 187.20	**<0.001** ^2^	23	39.64 ± 9.97	**<0.001** ^2^	23	77.40 ± 14.54	0.128 ^2^	23	0.52 ± 0.15	0.306 ^2^	21	9.95 ± 4.13	0.205 ^2^
50–59	570	507.49. ± 188.91	0.837 ^3^	399	43.48 ± 12.79	0.089 ^3^	399	77.61 ± 13.70	0.946 ^3^	399	0.58 ± 0.19	0.122 ^3^	393	11.97 ± 4.01	**0.041** ^3^
≥60	970	443.38 ± 191.50	**0.001** ^4^	752	37.28 ± 12.47	0.279 ^4^	752	75.92 ± 13.54	0.635 ^4^	752	0.52 ± 0.66	0.999 ^4^	723	12.07 ± 6.02	**0.033** ^4^
**Age in women**															
≤49	65	469.58 ± 176.10	**0.001** ^2^	9	38.96 ± 10.68	**<0.001** ^2^	9	80.92 ± 16.46	**0.007** ^2^	9	0.50 ± 0.16	**<0.001** ^2^	9	8.15 ± 3.61	0.105 ^2^
50–59	307	529.61 ± 179.76	**0.015** ^3^	211	46.50 ± 11.77	0.069 ^3^	211	78.59 ± 15.02	0.687 ^3^	211	0.61 ± 0.18	0.082 ^3^	209	11.69 ± 3.85	**0.019** ^3^
≥60	468	481.75 ± 194.71	0.608 ^4^	353	40.63 ± 12.28	0.656 ^4^	353	75.23 ± 11.46	0.331 ^4^	353	0.55 ± 0.18	0.402 ^4^	345	11.95 ± 6.12	**0.014** ^4^
**Age in men**															
≤49	61	540.02 ± 193.21	**<0.001** ^2^	14	40.07 ± 9.88	**<0.001** ^2^	14	75.13 ± 13.29	0.936 ^2^	14	0.54 ± 0.14	0.862 ^2^	12	11.30 ± 4.11	0.834 ^2^
50–59	263	481.67 ± 196.27	**0.037** ^3^	188	40.10 ± 13.07	0.995 ^3^	188	76.51 ± 11.98	0.712 ^3^	188	0.54 ± 0.20	0.944 ^3^	184	12.27 ± 4.17	0.446 ^3^
≥60	502	407.60 ± 181.48	**<0.001** ^4^	399	34.31 ± 11.88	**0.051** ^4^	399	76.54 ± 15.13	0.705 ^4^	399	0.50 ± 0.89	0.521 ^4^	378	12.17 ± 5.94	0.491 ^4^

*p*-value: ^1^ Student’s *t*-test for comparison of sex groups; ^2^ ANOVA for comparison of age groups; ^3^ Student’s *t*-test for comparison of young vs. middle-aged; ^4^ Student’s *t*-test for comparison of young vs. older-aged.

**Table 3 cancers-14-04563-t003:** Levels of serum sCD26 and DPP-IV activity according to colonoscopy findings.

Pathology	sCD26 N Mean ± SD (ng/mL) *p*-Value			DPP4 N Mean ± SD (ng/mL) *p*-Value			Total Protein N Mean ± SD (ng/mL) *p*-Value			Sp. Act (DPP4/Prot) N Mean ± SD (ng/mL) *p*-Value			sCD26/DPP4 Ratio N Mean ± SD (ng/mL) *p*-Value		
**No neoplasia**	1072	518.65 ± 183.93	**0.033** ^1^	671	43.45 ± 11.37	**0.014** ^1^	671	76.30 ± 12.60	0.469 ^1^	671	0.61 ± 0.69	0.338 ^1^	661	12.34 ± 5.09	0.190 ^1^
NCF	341	543.70 ± 186.88	-	188	45.36 ± 11.57	-	188	75.50 ± 11.81	-	188	0.71 ± 1.27	-	185	12.85 ± 4.72	-
hemorrhoids	116	496.55 ± 176.98	**0.018** ^2^	55	42.31 ± 12.69	0.094 ^2^	55	75.99 ± 11.21	0.783 ^2^	55	0.56 ± 0.17	0.391 ^2^	52	11.43 ± 4.11	0.050 ^2^
diverticula	96	496.76 ± 181.62	**0.029** ^2^	52	42.91 ± 9.63	0.165 ^2^	52	73.92 ± 9.98	0.380 ^2^	52	0.59 ± 0.14	0.488 ^2^	51	10.99 ± 3.41	**0.009** ^2^
polyps *	45	481.22 ± 140.90	**0.031** ^2^	14	37.17 ± 9.04	**0.010** ^2^	14	74.79 ± 7.81	0.823 ^2^	14	0.51 ± 0.15	0.554 ^2^	13	11.68 ± 2.64	0.377 ^2^
other ^†^	24	541.30 ± 190.95	0.952 ^2^	19	47.60 ± 12.93	0.426 ^2^	19	77.07 ± 11.95	0.581 ^2^	19	0.64 ± 0.24	0.811 ^2^	19	12.55 ± 5.75	0.793 ^2^
NAA	450	512.57 ± 185.82	**0.020** ^2^	343	42.69 ± 11.11	**0.010** ^2^	343	77.18 ± 13.71	0.157 ^2^	343	0.57 ± 0.17	**0.047** ^2^	341	12.41 ± 5.61	0.369 ^2^
**Advanced neoplasia**	594	381.85 ± 176.48	**<0.001** ^3^	503	34.08 ± 12.80	**<0.001** ^3^	503	76.82 ± 14.90	0.530 ^3^	503	0.45 ± 0.17	**<0.001** ^3^	476	11.51 ± 5.73	**0.010** ^3^
AA	345	419.95 ± 169.05	**<0.001** ^4^	261	38.13. ± 11.55	**<0.001** ^4^	261	77.91 ± 12.01	0.077 ^4^	261	0.50 ± 0.16	**0.010** ^4^	234	11.60 ± 4.53	**0.049** ^4^
CRC	249	329.05 ± 173.25	**<0.001** ^5^	242	29.72 ± 12.66	**<0.001** ^5^	242	75.65 ± 17.43	0.591 ^5^	242	0.40 ± 0.17	**<0.001** ^5^	242	11.42 ± 6.70	**0.029** ^5^

*p*-value: ^1^ ANOVA test for comparison of the 6 no neoplasia subgroups; ^2^ Student’s *t*-test for comparison of NCF vs. each of the NN subgroups; ^3^ Student’s *t*-test for comparison of no neoplasia vs. advanced neoplasia; ^4^ Student’s *t*-test for comparison of no neoplasia vs AA; ^5^ Student’s *t*-test for comparison of no neoplasia vs. CRC; NCF: no colorectal findings; * inflammatory and hyperplasic polyps; ^†^ others include angiodysplasia, rectitis and melanosis coli among others; NAA: non-advanced adenomas; AA: advanced adenomas; CRC: colorectal cancer. *p*-values in bold are statistically significant.

## Data Availability

Raw data can be obtained from the corresponding authors.

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
