# Peer review of "Evaluation of Blood Soluble CD26 as a Complementary Biomarker for Colorectal Cancer Screening Programs"

_cancers, 2022, doi:10.3390/cancers14194563_

Round 1

Reviewer 1 Report

sCD26, a soluble form of cell surface CD26, is one of the biomarkers discovered in CRC patients. In the manuscript, the authors studied whether the blood biomarker soluble CD26 and DPP4 activity could help reduce false positives in early diagnosis of colorectal cancer in combination with FIT. By analyzing a large data cohort, they found that combined information of sCD26, DPP4 and their ratio identified almost advanced adenomas and CRC cases in FIT cohort with half false positives than FIT. They also proposed a sequential testing strategy for FIT positive individuals. The proposal could reduce the number of colonoscopies, which is convenient for the health system and the patients. The manuscript is well organized and well written. The figures and tables are well organized. The results are fully discussed, and the conclusions are consistent with the data they presented. There are also a couple of concerns that need to be addressed. 

  1. In Figure 2, the patients tested FIT positive and sCD26/DPP4 positive will be submitted to colonoscopy. How about the patients with FIT-sCD26/DPP4+ or FIT+ sCD26/DPP4-? The authors should clearly indicate where these patients go in the figure. 
  2. The screening strategy the authors proposed could reduce the number of colonoscopies, and at the same time it also increased the costs and the labor of screening by testing sCD26 and DPP4. Is it still a benefit to the health system and patients? 

Reviewer 2 Report

Thanks for submitting the manuscript to Cancers. The authors included the serologic test of sCD26, DPP4 in combination with FIT to decrease the false positivity of FIT in colorectal cancer screening in the Spanish population. By adding the serologic tests, the authors could decrease the false positive cases by about half of the cases. 

The strength of the study;

The authors introduced promising serum biomarkers to decrease the false positivity of FIT. The cut-off levels of each test and the ratio were proposed. The authors also proposed a new strategy to decrease the false-positive cases or colonoscopy based on the study findings.

The weakness of the study;

It might be an editing problem, I think. all the Tables need re-editing, the rows are not well aligned. 

The numbers in the study cohorts are complicated. Because this study starts from the FIT tests, the total number of patients should be 917 patients rather than 1703 patients. The study aims to add the serologic test to complement the FIT in CRC screening. 

The sCD 26 and DDP4 test has many limitations, as the authors described in the manuscript. To determine the cut-off level of both serologic makers, I would like to suggest using the ROC curves rather than the Student's T-test or ANOVA analysis.

Moreover, screening test needs definite economic concerns. Please describe this issue in the discussion. How much money can we save based on the new strategy you proposed in the manuscript?  

The manuscript is hard to read or follow because the authors described too much information from the study results. 

I found some mis-spelling or abbreviations in lines 222, 224 (AN), and 346 (CSC).  
